# Plasma Osteopontin Levels Is Associated with Biochemical Markers of Kidney Injury in Patients with Leptospirosis

**DOI:** 10.3390/diagnostics10070439

**Published:** 2020-06-29

**Authors:** Haorile Chagan-Yasutan, Firmanto Hanan, Toshiro Niki, Gaowa Bai, Yugo Ashino, Shinichi Egawa, Elizabeth Freda O. Telan, Toshio Hattori

**Affiliations:** 1Department of Health Science and Social Welfare, Kibi International University, 8 Igamachi, Takahashi 716-8508, Japan; haorile@foxmail.com (H.C.-Y.); gaowabai@kiui.ac.jp (G.B.); 2Mongolian Psychosomatic Medicine Department, International Mongolian Medicine Hospital of Inner Mongolia, Hohhot 010065, China; 3Disease Control and Prevention Division of Banyuasin District Health Office, Pangkalan Balai, South Sumatra 30753, Indonesia; 009fireman@gmail.com; 4Department of Immunology, Faculty of Medicine, Kagawa University, Kita-gun 761-0793, Japan; niki@med.kagawa-u.ac.jp; 5Department of Respiratory Medicine, Sendai City Hospital, Miyagi 982-8502, Japan; ya82@yahoo.co.jp; 6International Research Institute of Disaster Science, Tohoku University, Sendai 980-8575, Japan; egawas2@irides.tohoku.ac.jp; 7National Reference Laboratory for HIV/AIDS, Hepatitis, and other STDs, STD/AIDS Cooperative Central Laboratory, Manila 1003, Philippines; betelan@yahoo.com

**Keywords:** leptospirosis, biomarker, osteopontin, galectin-9, cystatin C, clusterin, creatinine

## Abstract

Leptospirosis becomes severe, with a fatality rate of >10%, and manifests as severe lung injury accompanied by acute kidney injury. Using urine and blood samples of 112 patients with leptospirosis, osteopontin (OPN), galectin-9 (Gal-9) and other kidney-related biomarkers were measured to understand the pathological and diagnostic roles of OPN and Gal-9 in leptospirosis. Plasma levels of full-length (FL)-OPN (pFL-OPN) (*p* < 0.0001), pFL-Gal-9(*p* < 0.0001) and thrombin-cleaved OPN (*p* < 0.01) were significantly higher in patients with leptospirosis than in healthy controls (*n* = 30), as were levels of several indicators of renal toxicity: serum cystatin C (*p* < 0.0001), urine *N*-acetyl-β-glucosaminidase (NAG)/creatinine (*p* < 0.05), and urine clusterin/creatinine (*p* < 0.05). pFL-Gal-9 levels were negatively correlated with pFL-OPN levels (*r* = −0.24, *p* < 0.05). pFL-OPN levels were positively correlated with serum cystatin C (*r* = 0.41, *p* < 0.0001), urine NAG/creatinine (*r* = 0.35, *p* < 0.001), urine clusterin/creatinine (*r* = 0.33, *p* < 0.01), and urine cystatin C/creatinine (*r* = 0.33, *p* < 0.05) levels. In a group of patients with abnormally high creatinine levels, significantly higher levels of serum cystatin C (*p* < 0.0001) and pFL-OPN (*p* < 0.001) were observed. Our results demonstrate that pFL-OPN reflect kidney injury among patients with leptospirosis.

## 1. Introduction

Leptospirosis is a neglected zoonotic disease with a global distribution, endemic mainly in countries with humid subtropical or tropical climates, and has epidemic potential [1]. There are an estimated 1.03 million cases per year, resulting in 2.9 million disability-adjusted life years, with the highest burden falling on resource-poor tropical countries [2]. About 5–10% of patients with leptospirosis can potentially develop a severe form of the disease, with a fatality rate of >10% for Weil’s disease and up to 70% for leptospirosis pulmonary hemorrhage syndrome [3]. The most severe form of the disease (Weil’s disease) manifests as severe lung injury (diffuse alveolar hemorrhage, pulmonary edema, acute respiratory distress syndrome, or a combination of these features) accompanied by acute kidney injury (AKI) [4,5]. The incidence of AKI in leptospirosis varies from 40 to 60%, depending on the definition, and oliguria is associated with a worse outcome [6].

Leptospirosis may be misdiagnosed as malaria, viral hepatitis, influenza, dengue fever (DF), rickettsiosis, typhoid fever, or melioidosis. It was shown that DF and leptospirosis epidemics are correlated not only with rainfall but also with relative humidity and temperature in the Philippines. The peak occurrence of leptospirosis preceded that of DF by only one month [7].

Biomarkers that can support early detection, monitor disease progression, and follow-up prognosis are valuable in the clinical diagnosis and treatment of patients with leptospirosis. Given the importance of urinary markers in kidney injury associated with leptospirosis, urine defensin alpha 1, urine neutrophil gelatinase-associated lipocalin, and urine *N*-acetyl-β-d-glucosidase (uNAG)—markers of AKI and tubular dysfunction—are elevated in leptospirosis patients and reflect kidney damage [8]. A study conducted in Thailand also reported that leptospirosis patients had acute renal failure with increased uNAG and urine neutrophil gelatinase-associated lipocalin levels [9,10].

It was proposed that dramatic imbalances in cytokine production in leptospirosis might play a considerable role in the development of severe leptospirosis. These data suggest that tumor necrosis factor-α and interleukin-1β are master conductors of the awry inflammatory response and the subsequent cytokine storm-induced sepsis observed in severe leptospirosis [11]. We have studied full-length osteopontin (FL-OPN) [12] and full-length galectin-9 (FL-Gal-9) [13] in dengue patients and showed that the former directly correlated with D-dimer and ferritin levels, and the latter tracked viral load, and were associated with multiple cytokines and chemokines. It is also known that dengue patients suffer from cytokine storm like leptospirosis. We chose these two matricellular proteins because OPN is known to enhance the Th1-mediated inflammatory response and plays a key role in apoptosis [14], while an enhanced Th2 cell-mediated immune response was observed in the kidneys of nephritic mice after a 7-day injection of Gal-9 [15]. It is also known that Gal-9 has apoptosis inducing activity [16]. It is already known that the outer membrane of Gram-negative bacteria, lipopolysaccharide, can increase the levels of Gal-9 in bacterial infection [17]. The leptospiral outer membrane constituents (lipoprotein 32 and *Leptospira* surface adhesin) were shown to activate macrophages through the toll-like receptor (TLR) pathway and establish the predominant signaling component for macrophages through this pathway [18]. A recent study reported that the activation of the MyD88 pathway by TLR2, TLR5, and TLR7/8 agonists or interleukin-1 induces high levels of OPN in human dendritic cells. TLR2 agonists were the strongest OPN inducers, and OPN production was highly stimulated by TLR2-triggering bacteria [19].

This study aimed to assess the level of FL-Gal-9 and FL-OPN in patients with leptospirosis and to evaluate their correlation with markers of renal toxicity for the first time.

## 2. Materials and Methods

### 2.1. Study Subjects

A retrospective case control study was performed in febrile patients with clinical suspicion of leptospirosis (WHO, 2003), who were admitted to San Lazaro Hospital (SLH, Manila, Philippines) after the August 2012 flood. The following diagnostic tools using serum or urine samples were employed, such as the microscopic agglutination test [20], immunochromatographic assay (Standard Diagnostics, Yongin, Korea), enzyme-linked immunosorbent assay (ELISA; Diagnostic Automation, Calabasas, CA, USA), loop-mediated isothermal amplification, and real-time PCR [21]. Among the patients admitted, 112 patients with leptospirosis were confirmed and enrolled in this study. Thirty healthy controls (HCs) were volunteers from the employees of SLH.

EDTA-plasma, serum, and spot urine were obtained via centrifugation and aliquoted into CryoTubes for storage at −80 °C until further utilization. All the samples were collected at the time of admission. Among the HCs, spot urine and serum samples were only available for ten individuals.

### 2.2. Inflammatory Markers

Plasma and urine FL-OPN (pFL-OPN and uFL-OPN) concentrations were determined using commercially available ELISA kits (Human opn assay kit; Immuno-Biological Laboratories, Takasaki, Japan). The levels of plasma and urine tr-OPN (ptr-OPN and utr-OPN) were measured using Human OPN N-half Assay Kits (Immuno-Biological Laboratories), which specifically measure the FL-OPN cleaved by thrombin. Both values of FL-OPN and tr-OPN were expressed as pmol/L (pM) [12]. The plasma concentration of FL-Gal-9 (pFL-Gal-9) was quantified via ELISA (Galpharma Co. Ltd, Takamatsu, Japan) as described previously [13]. The kit was found to be specific to FL-Gal-9 [22]. pFL-Gal-9 values were expressed in pg/mL.

The levels of uNAG, serum creatinine (sCr), and serum and urinary cystatin C (sCyC and uCyC) were measured at Special Reference Laboratories, Hachiouji, Japan, and the levels were expressed as U/L, mg/dL, and mg/L, respectively. Kidney dysfunction was defined according to the sCr (normal level: men, 0.61–1.04 mg/dL; women, 0.47–0.79 mg/dL). sCr levels higher than normal were marked as positive and kidney injury. In addition, the urine levels of human kidney biomarkers of toxicity (albumin, β-2-microglobulin, clusterin [uCLU], and uCyC) were measured using MILLIPLEX^®^ MAP Human Kidney Toxicity Magnetic Bead Panel 4 (Merck Millipore, Billerica, MA, USA). The levels of these biomarkers were expressed in pg/mL. The levels of urine albumin and β-2-microglobulin were not included in the analysis because some values were out of range.

All urinary marker levels were adjusted relative to the levels of urinary Cr (uCr), which were determined using the Cr parameter assay kit (R&D Systems, Minneapolis, MN, USA) according to the manufacturer’s instructions [8].

### 2.3. Dipstick Analysis

A dipstick kit (Eiken Chemical Co., Tokyo, Japan) was used to determine the presence of red blood cells, leukocytes, albumin, and glucose in urine.

### 2.4. Statistical Analysis

The data are expressed as medians and ranges unless indicated otherwise. The levels of biomarkers in leptospirosis patients and HCs were compared using the Mann–Whitney *U* test. The correlation between inflammation biomarkers and the levels of biomarkers of human kidney toxicity in leptospirosis patients was assessed using the Spearman rank correlation coefficient. The above computations were performed using GraphPad PRISM version 8 (GraphPad Software, San Diego, CA, USA). Receiver operating characteristic (ROC) curves were generated, and the AUC was calculated to determine the trade-off between the sensitivity and specificity for distinguishing leptospirosis cases from HCs using Medcalc statistical software version 19 (Ostend, Belgium). Analysis items with *p* ≤ 0.05 were considered as being statistically significant.

### 2.5. Ethics Statement

The study was conducted in accordance with the tenets of the Declaration of Helsinki. The study protocol was approved by the ethics committees of San Lazaro Hospital, Manila, Philippines (2011-08-010) and Tohoku University Hospital, Sendai, Japan (2012-1-170). Written informed consent was obtained from all study participants prior to inclusion in the study.

## 3. Results

### 3.1. Characteristics of Study Participants

A total of 112 leptospirosis patients (94 males and 18 females) and 30 healthy controls (HCs) (9 males and 21 females) were enrolled in this study. The median ages of leptospirosis patients and HCs were 30 (range, 12–67) years and 28.50 (range, 22–59) years, as reported previously [8,21]. The participating leptospirosis patients and HCs had similar distributions of age (*p* = 0.6).

### 3.2. Comparison of Biomarkers among Leptospirosis Patients and Healthy Controls

Compared with HCs, leptospirosis patients had significantly higher levels of plasma (p)FL-OPN (*p* < 0.0001), plasma thrombin-cleaved (ptr)-OPN (*p* < 0.01), pFL-Gal-9 (*p* < 0.0001), serum creatinine (sCr) (*p* < 0.0001), and serum cystatin C (sCyC) (*p* < 0.0001) (Figure 1). Among urinary biomarkers, the level of urine clusterin (uCLU)/Cr (*p* < 0.05) was significantly higher in leptospirosis patients. No significant differences in uFL-OPN/Cr, utr-OPN/Cr, and uCyC/Cr levels were observed between the groups (Figure 1).

Receiver operating characteristic (ROC) curve analysis was used to discriminate leptospirosis patients from HCs. All markers had area under the ROC curve (AUC) values of >0.5. The result of ROC analysis of plasma samples (Figure 2A) shows that pFL-Gal-9 levels had the greatest ability to discriminate leptospirosis patients from HCs based on the AUC (0.953), followed by sCyC (0.934), sCr (0.892), pFL-OPN (0.875), and ptr-OPN (0.632) (Figure 2A). Both the sensitivity and specificity of pFL-Gal-9 were more or equal with 90 percent associated with the highest Youden index (Table 1).

In urine samples, the urinary *N*-actyl-β-d-glucosaminidase (uNAG)/Cr levels had the greatest ability to discriminate the leptospirosis group from HCs based on the AUC (0.849), followed by uCLU/Cr (0.731), utr-OPN/Cr (0.667), and uFL-OPN (0.619) (Figure 2B). It is of note that the specificity of uTr-OPN is the highest (100%), but its sensitivity is low (34.3%) (Table 1). Based on these results, the levels of pFL-Gal-9 and sCyC were more able to discriminate leptospirosis patients from HCs compared with urine markers.

### 3.3. Association between Leptospirosis Marker Levels and Dipstick Parameters

The associations between the above-mentioned urinary markers and urinary dipstick parameters were analyzed. pFL-OPN levels were significantly higher in the leukocyte-positive group (*p* < 0.01) and glucose-positive group (*p* < 0.05) (Figure 3), while ptr-OPN levels were significantly higher only among patients with hematuria (*p* < 0.001). Among urine samples, uFL-OPN/Cr was significantly higher in the urine albumin-positive group (Figure 3A,B). uCyC/Cr levels were higher in the leukocyte-positive, glucose-positive, and proteinuria groups as compared to the corresponding negative groups (*p* < 0.01, *p* < 0.05, and *p* < 0.001, respectively) (Figure 3C). This may indicate that uCyC/Cr reflects the state of local infection or urinary tract dysfunction as compared with sCyC. pFL-Gal-9 levels were significantly higher only in the albuminuria group (*p* < 0.05) (Figure 3D). Furthermore, uCLU/Cr levels were significantly higher in groups that tested positive to all the parameters (hematuria-positive, leukocyte-positive, glucose-positive, and proteinuria groups) (*p* < 0.001, *p* < 0.01, *p* < 0.05, and *p* < 0.0001, respectively) (Figure 3E). Therefore, uCLU/Cr may well reflected kidney injury in leptospirosis patients, such as local infection or urinary tract dysfunction (Figure 3E).

### 3.4. Correlation between Biomarker Levels in Leptospirosis Patients

pFL-Gal-9 levels showed an inverse correlation with pFL-OPN levels (*r* = −0.24, *p* < 0.05) but showed no correlation with other markers (Figure 4), while pFL-OPN levels correlated significantly with the levels of sCyC (*r* = 0.41, *p* < 0.0001), utr-OPN/Cr (*r* = 0.23, *p* < 0.05), uNAG/Cr (*r* = 0.35, *p* < 0.001), uCLU/Cr (*r* = 0.33, *p* < 0.05), uCyC/Cr (*r* = 0.33, *p* < 0.05), and sCr (*r* = 0.28, *p* < 0.01), indicating that OPN was involved in kidney injury. Similarly to pFL-OPN, ptr-OPN was correlated with sCyC, uNAG/Cr, uCLU/Cr, uCyC/Cr, and sCr levels.

Moreover, 39% of the enrolled leptospirosis patients tested positive for sCr. The sCr-positive and -negative groups were studied to assess whether they expressed Gal-9 or OPN differently. Compared with the sCr-negative group, the sCr-positive group showed significantly higher levels of sCyC (*p* < 0.0001) and pFL-OPN, but not pFL-Gal-9, suggesting that OPN played a role in kidney disease (Figure 5).

## 4. Discussion

For the first time, our study showed that the levels of FL-OPN, tr-OPN, and FL-Gal-9 were significantly increased in the plasma of patients with leptospirosis. ROC analysis clearly showed that pFL-Gal-9 levels had the highest AUC value. This finding may indicate that pFL-Gal-9 levels could reflect the severity of leptospirosis as reported previously in dengue [13] and malaria [23], in which cytokinemia is also frequently seen, as in leptospirosis [11]. A recent proposal of galectin as checkpoints of the immune system [24,25] may give the possibility that sudden degradation of Gal-9 may lead to uncontrolled systemic inflammation, such as a cytokine storm. However, significant differences in pFL-Gal-9 levels on urine dipstick examination were only seen in the albuminuria group, and no correlation was found with kidney markers such as uNAG/Cr, sCr and sCyC, suggesting that FL-Gal-9 does not reflect kidney injury (Figure 2A, Figure 3D, Figure 4 and Figure 5). Serum Gal-9 levels were reportedly elevated in chronic kidney diseases with diabetes but were not correlated with albuminuria [26]. It is also known that galectin is induced in kidneys during the early phase of leptospirosis infection in mice [27]. It should also be noted that administration of Gal-9 ameliorates glomerulonephritis in mice, though it is not known whether the increased Gal-9 could ameliorate kidney injury in humans or not [15].

The correlation of pFL-OPN level with uNAG/Cr, sCyC, uCyC, and uCLU/Cr values indicate that OPN might be involved in tubular dysfunction and is released by infiltrating inflammatory macrophages in tubules [28]. There was a significant increase in uNAG/Cr (*p* < 0.05) levels in leptospirosis patients as described previously [8]. An increased uNAG level is a highly specific marker of proximal tubular disease [29]. A study conducted in Thailand also reported that 45% of leptospirosis patients had acute renal failure with increased levels of uNAG and β2-microglobulin, which indicate proximal tubular dysfunction [9]. The obstructed kidneys of OPN knockout mice exhibit increased levels of tubular cell apoptosis compared to wild-type mice, suggesting that OPN is capable of providing survival signals to tubular epithelial cells in vivo [30]. The binding of OPN to the alpha-v beta-3 integrin of endothelial cells activates the pro-survival transcription factor nuclear factor-kappa B and protects endothelial cells from undergoing apoptosis [31]. It is interesting to find a correlation between pFL-OPN and uCLU/Cr because clusterin has been suggested to play a vital role in the injury of proximal tubular cells and attenuate cell death [32]. Apparent higher correlations between uNAG, uCyC/Cr and uCLU/Cr, common biomarkers of tubular injury, than those with pFL-OPN, and a lack of correlation of uFL-OPN/Cr with tubular injury biomarkers, imply more detailed analysis would be necessary. In lupus nephritis, urinary levels of adiponectin and osteopontin predict chronic kidney damage with similar accuracy as the glomerular filtration rate (GFR) using a different ELISA which detects both FL-OPN and its cleaved form [12,33]. It is also known that mononuclear-macrophages but not neutrophils act as major infiltrating anti-leptospiral phagocytes during leptospirosis, therefore OPN could reflect infiltrated macrophage and disease severity [28]. Leptospira exposure was proposed to play a role in chronic kidney diseases (CKD) as a cause of primary kidney disease [34]; the development of CKD maybe detected early by periodic measuring OPN.

Although OPN may play a role in tubular dysfunction, one cannot exclude the possibility that OPN may have other functions in kidneys or other tissues [14]. The moderate association observed between plasma pFL-OPN level and sCyC supports the idea, because CyC, a non-glycosylated protein with a low molecular weight of 13 kDa and continuously produced by all nucleated cells, is directly filtered from blood in the glomerulus, and its serum levels are an ideal estimator of the glomerular filtration rate [35]. It was reported that among patients admitted to the emergency department, plasma OPN level correlated well with sCr level and could be an independent predictor of sepsis; however, no difference was found between patients with and without AKI using an ELISA kit whose epitope was not disclosed but was found to react with both full-length and cleaved form of OPN [12,36,37].

No significant correlation was found between pFL-OPN and ptr-OPN; however, uFL-OPN/Cr was correlated with utr-OPN/Cr. Furthermore, it is noteworthy that pFL-OPN and ptr-OPN showed significant correlations with uNAG/Cr, pCyC, uCyC/Cr, and uCLU/Cr. Moreover, the levels of ptr-OPN were found to be higher in the hematuria group, as determined by urine dipstick analysis. These findings indicate the association of both FL-OPN and tr-OPN in kidney injury. In another study, utr-OPN was found in the urine but not in the plasma of patients with lupus nephritis, and utr-OPN was proposed to play a role in lupus nephritis [38]. These findings indicate that the presence of tr-OPN in not only the urine but also the plasma of leptospirosis patients may reflect activation of proteases in plasma, because tr-OPN is known to be generated by thrombin cleavage [39]. The cleavage might occur by secreted proteases not only by inflammatory tissues but also by *Leptospira* sp [40].

In this study, we assessed Gal-9 and OPN for the first time in patients with leptospirosis and found elevated levels of pFL-Gal-9 and pFL-OPN and ptr-OPN. OPN is closely involved in kidney disease; however, there is little evidence of the involvement of Gal-9 in kidney disease. Furthermore, pFL-GAL-9 and pFL-OPN were inversely correlated. The reasons for the inverse association are unclear; however, bacterial infections seem to induce the expression of Gal-9, as evident from the increase in Gal-9 levels in bacterially infected periodontal ligament cells [41]. Secreted Gal-9, in turn, downregulates TL2 and TL4 and suppresses the release of inflammatory cytokines [42]. This suppressive effect may downregulate OPN and lead to an inverse relationship between them. It is also possible that OPN produced in the kidney may attract inflammatory cells such as monocytes and neutrophils [8,43] and may cause a relative decrease in systemic inflammatory cells, leading to decreased levels of pFL-Gal-9. In addition, a slight elevation of pFL-Gal-9 levels was found only in the albuminuria group on dipstick examination, and we could not confirm that pFL-Gal-9 reflects kidney injury. As described above, administration of Gal-9 was known to ameliorate glomerulonephritis by inhibiting Th1 response in mice, therefore it is possible that produced Gal-9 may be beneficial for kidney injury in leptospirosis [15]. The large AUC for pFL-Gal-9 suggests that pFL-Gal-9 levels may reflect systemic or other inflammation in patients with leptospirosis, as observed in dengue and malaria [13,23]. It was recently reported that derivatives of brefelamide, an aromatic amide isolated from fruiting bodies of *Dictyostelium brefeldianum* and *Dictyostelium giganteum*, inhibits synthesis of OPN and other inflammatory molecules including Gal-9. These molecules could be candidates for therapy in acute kidney injury in severe forms of leptospirosis [44].

Our study has several limitations. First, spot urine samples were used. Second, low numbers of controls and few urine samples of HCs were available for analysis in the present study. Third, since leptospirosis itself is mostly found among men in the outdoor or flood environment, the patients enrolled in this study were predominantly male. Fourth, clinical data regarding estimated glomerular filtration rate, diabetes, hypertension, heart failure, underlying chronic kidney disease, C-reactive protein, mean arterial pressure, and hemoglobin for these patients were not collected in this study. Finally, longitudinal studies with larger sample sizes should be conducted to validate the results.

In conclusion, our study presented that pFL-Gal-9, pFL-OPN, and ptrOPN and other kidney toxicity markers are elevated in leptospirosis patients. pFL-Gal-9 gave the highest AUC values as compared with healthy controls. pFL-OPN levels were associated with those of kidney toxicity markers indicating diagnostic value for AKI and may be useful for monitoring the occurrence of CKD in leptospirosis.

## Figures and Tables

**Figure 1 diagnostics-10-00439-f001:**
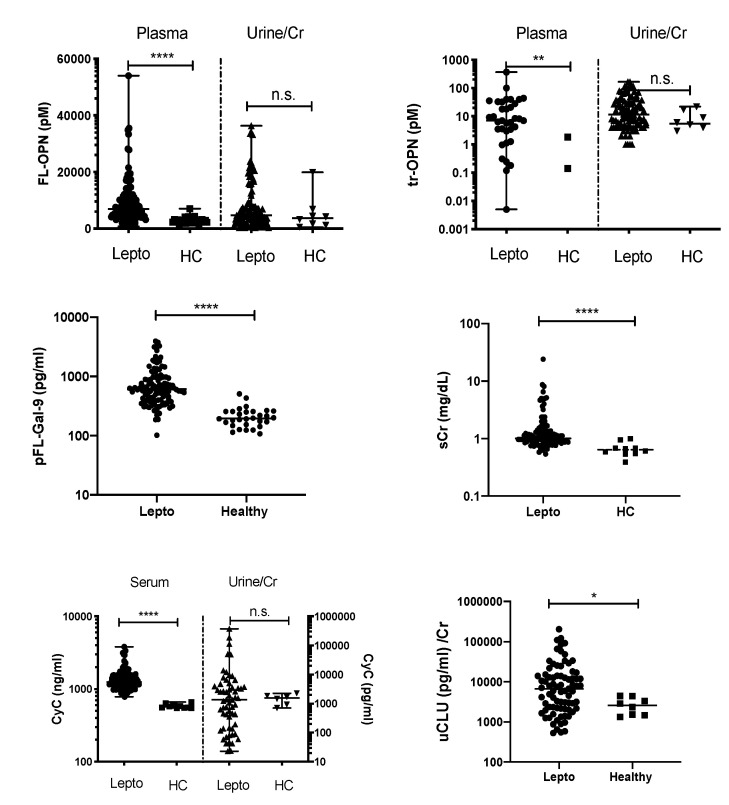
The levels of biomarkers in the plasma (p), serum (s), and urine (u) were compared between patients with confirmed leptospirosis and healthy controls. *p* value is written as: ns, not significant; * *p* < 0.05, ** *p* < 0.01, **** *p* < 0.0001. FL-OPN, full-length osteopontin; tr-OPN, thrombin-cleaved osteopontin; CyC, cystatin C; FL-Gal-9, full-length galectin-9; CLU, clusterin; Cr, creatinine.

**Figure 2 diagnostics-10-00439-f002:**
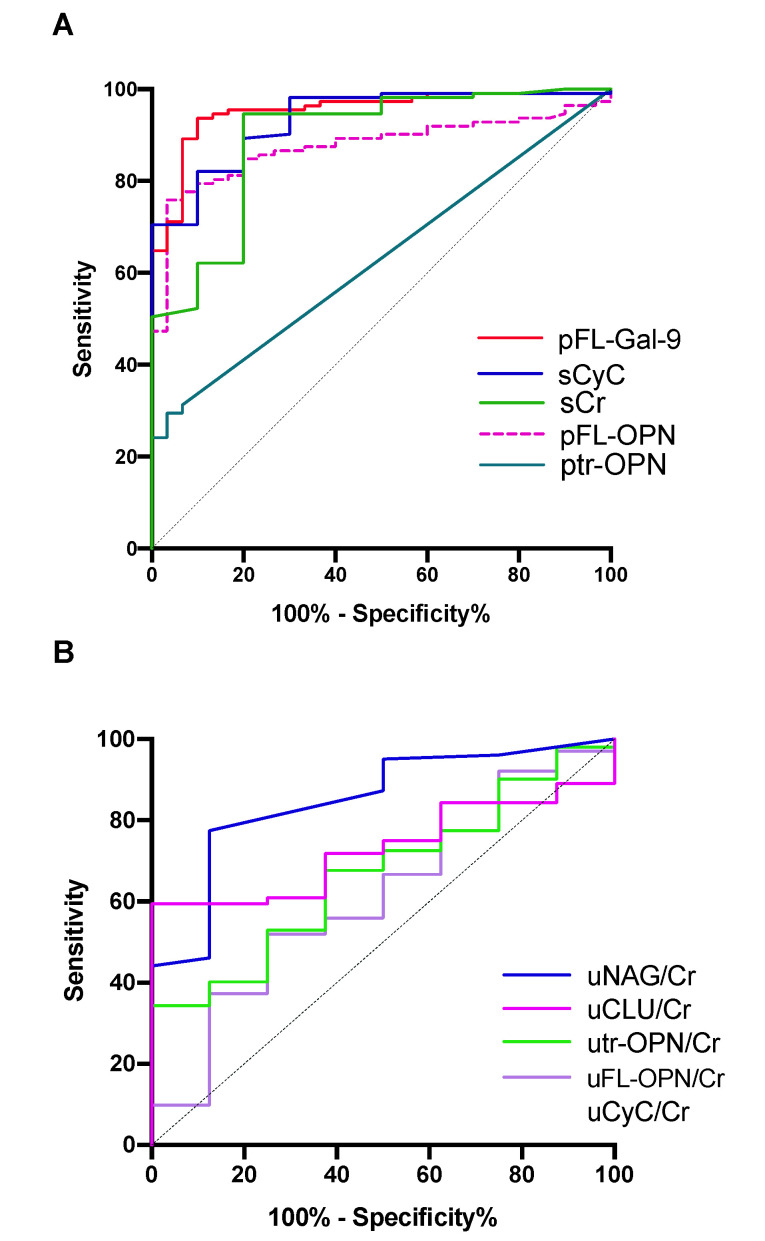
Receiver operating characteristic (ROC) curve analysis to discriminate leptospirosis patients from healthy controls. (**A**) Markers measured in plasma or serum; (**B**) markers measured in urine samples. Higher area under the ROC curve (AUC) values indicate a better ability to diagnose leptospirosis patients. FL-Gal-9, full-length galectin-9; CyC, cystatin C; Cr, creatinine; FL-OPN, full-length osteopontin; tr-OPN, thrombin-cleaved osteopontin; NAG, *N*-acetyl-β-d-glucosidase; CLU, clusterin; p, plasma; u, urine.

**Figure 3 diagnostics-10-00439-f003:**
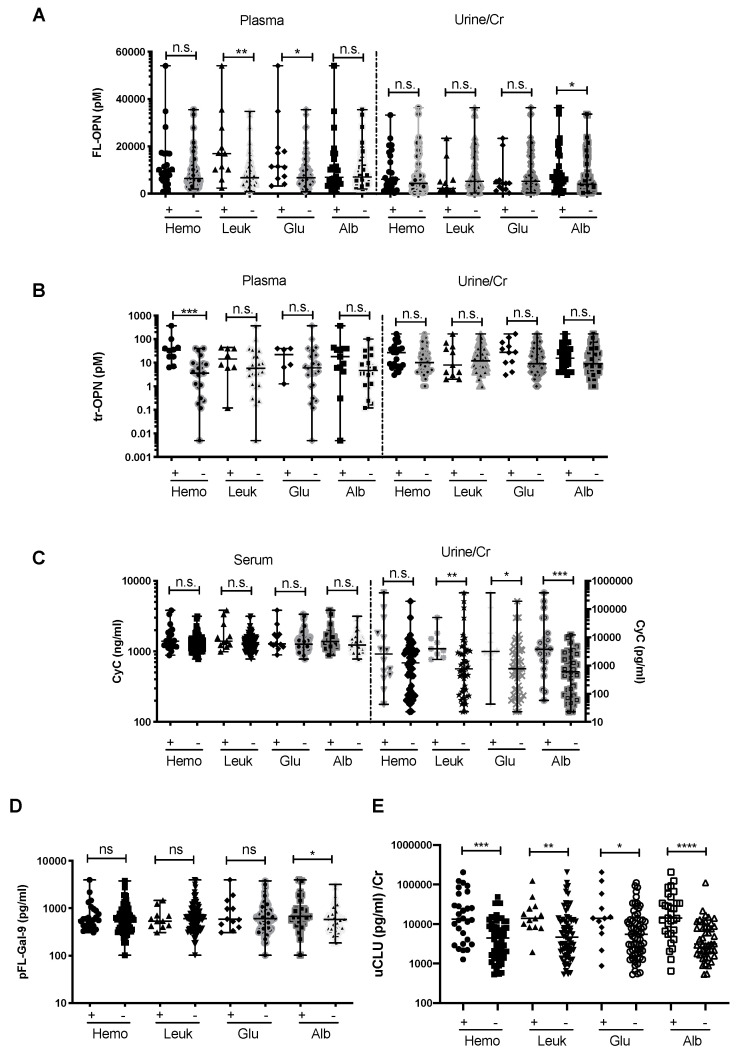
Plasma and urinary markers of FL-OPN (**A**), tr-OPN (**B**), Cyc (**C**), plasma pFL-Gal-9 (**D**), and uCLU/Cr (**E**) in hematuria (Hemo)-, leukocyte (Leuk)-, glucose (Glu)-, and albumin (Alb)-positive or -negative groups among leptospirosis patients. *p* value is written as: ns, no significant; * *p* < 0.05, ** *p* < 0.01, *** *p* < 0.001, **** *p* < 0.0001. FL-OPN, full length of osteopontin; tr-OPN, thrombin –cleaved osteopontin; CyC, cystatin C; FL-Gal-9, full length of galectin-9; CLU, clusterin; Cr, creatinine.

**Figure 4 diagnostics-10-00439-f004:**
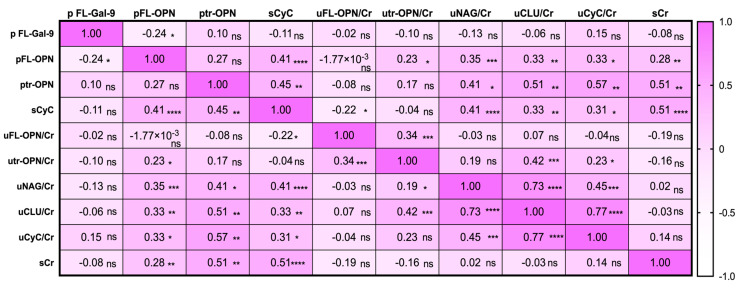
Heatmap of correlation matrix of plasma (p), serum (s) or urinary (u) markers in leptospirosis patients. The correlation was measured by Spearman *t* test. The correlation *r* value is written in each well and displayed as colors ranging from white to pink as shown in legend key. *p* value is written as: ns, no significant; * *p* < 0.05, ** *p* < 0.01, *** *p* < 0.001, **** *p* < 0.0001. FL-Gal-9, galectin-9; FL-OPN, full length of osteopontin; tr-OPN, thrombin –cleaved osteopontin; NAG, *N*-acetyl-β-d-glucosidase, CLU, clusterin; CyC, cystatin C; Cr, creatinine.

**Figure 5 diagnostics-10-00439-f005:**
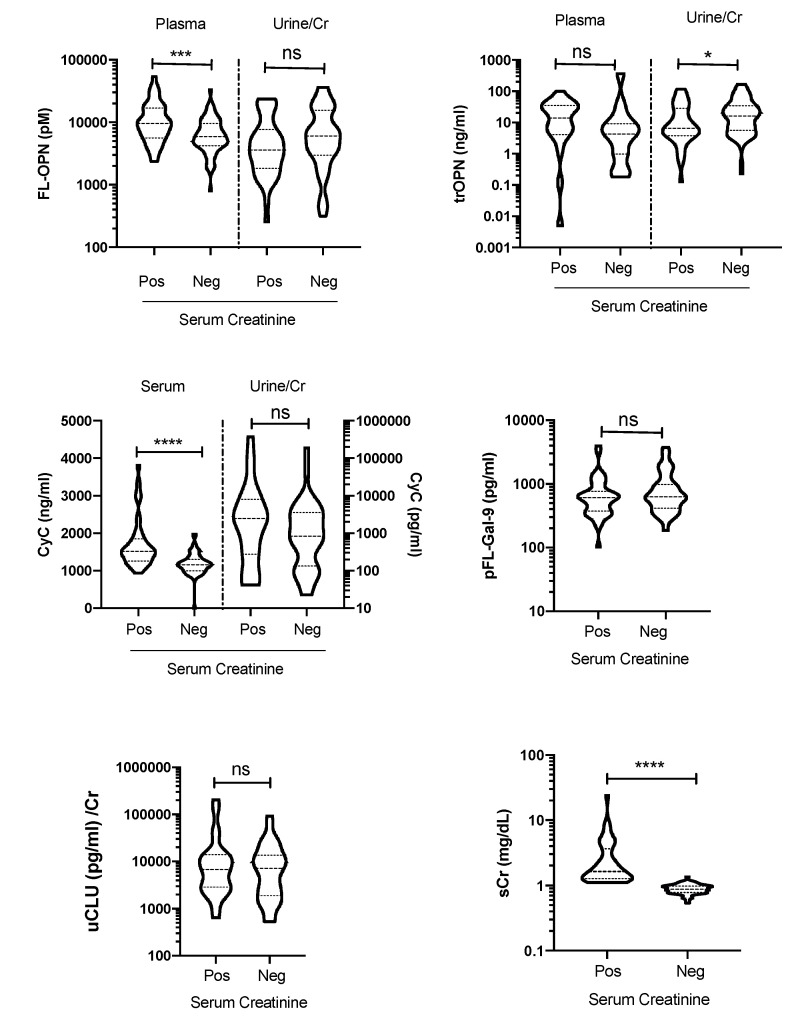
The levels of biomarkers in the plasma (p), serum (s), and urine (u) were assessed in the serum creatinine-positive (Pos) and -negative (Neg) groups. *p* value is written as: ns, no significant; * *p* < 0.05, *** *p* < 0.001, **** *p* < 0.0001. FL-OPN, full-length osteopontin; tr-OPN, thrombin-cleaved osteopontin; CyC, cystatin C; FL-Gal-9, full-length galectin-9; CLU, clusterin; Cr, creatinine.

**Table 1 diagnostics-10-00439-t001:** Summary of ROC curve analysis.

	AUC	*p* Value	Youden Index	Criterion Value	Sensitivity (%)/95% CI	Specificity (%)/95% CI
**pFL-Gal-9**	0.953	****	0.837	>286	93.7 (87.4~97.4)	90.0 (73.5~97.9)
**sCyC**	0.934	****	0.721	>1010	82.1 (73.8~88.7)	90.0 (55.5~99.7)
**sCr**	0.892	****	0.746	>0.68	94.6 (88.6~98.0)	80.0 (44.4~97.5)
**pFL-OPN**	0.875	****	0.726	>277	75.9 (66.9~83.5)	96.7 (82.8~99.9)
**ptr-OPN**	0.630	****	0.261	>0.14	29.5 (21.2~38.8)	96.7 (82.8~99.9)
**uNAG/Cr**	0.849	****	0.650	>0.04	77.5 (68.1~85.1)	87.5 (47.3~99.7)
**uCLU/Cr**	0.731	***	0.594	>4456	59.4 (46.4~71.5)	100.0 (63.1~100)
**utr-OPN/Cr**	0.667	*p* = 0.05	0.343	>21.9	34.3 (25.2~44.4)	100 (63.1~100)
**uFL-OPN/Cr**	0.619	*p* = 0.27	0.270	>285	52.0 (41.8~62.0)	75.0 (34.9~96.8)
**uCyC/Cr**	0.511	*p* = 0.87	0.424	≤2251	57.6 (44.1~70.4)	0 (0.0~45.9)

*p* value is written as: *** *p* < 0.001, **** *p* < 0.0001. FL-Gal-9, full-length galectin-9; CyC, cystatin C; Cr, creatinine; FL-OPN, full-length osteopontin; tr-OPN, thrombin-cleaved osteopontin; NAG, *N*-acetyl-β-d-glucosidase; CLU, clusterin; *p*, plasma; u, urine.

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
