# Peer review of "Plasma Osteopontin Levels Is Associated with Biochemical Markers of Kidney Injury in Patients with Leptospirosis"

_diagnostics, 2020, doi:10.3390/diagnostics10070439_

Round 1
Reviewer 1 Report
In the manuscript entitled “Plasma osteopontin levels but not galestin levels reflect kidney injury in patients with Leptospirosis” Chagan-Yasutan et al. showed significant high plasma levels of full-length galestin (pFL-Gal-9), osteopontin (pFL-OPN) and thrombin-cleaved OPN in patients with leptospirosis as compared to healthy controls. Plasma FL-OPN also significantly correlated with biochemical markers of acute renal tubular injury. The authors conducted a ROC analysis to suggest the utility of different plasma and urine molecules as a biomarker in leptospirosis patients. This is potentially interesting manuscript lacks several important points. The manuscript is required a major correction.
Major comments:
- The title is a little confusing since it could suggest that pFL-Gal-9 cannot be used as a biomarker of kidney injury in leptospirosis patients. The authors evaluated pFL-Gal-9 at only one-time point during infection. With the current results, the authors cannot suggest that pFL-Gal-9 is not a relevant biomarker of early kidney injury in leptospirosis, since previous studies in mice indicate that pFL-Gal-9 is elevated during acute kidney injury. Suggestion: Plasma Osteopontin Levels is associated with biochemical markers of Kidney Injury in Patients with Leptospirosis
- The rational to use osteoponin and galestin as biomarkers of kidney injury is not clear, especially because more common biomarkers of tubular injury (uNAG, CyC, sCr and uCLU) can well detect it. Morover correlations between uCyC/CRE, uCLU/Cr and uNAG/Cr are the highest in the st of markers (Fig.4).
- Provide information about the percentage of patients with leptospirosis that develop acute tubular disease and acute kidney failure.
- Matherial and methods: Study subjects. It is not clear the type of study design that was used: Case-control study, cross-sectional study, age-matched case-control, etc. When the samples were collected: time 0, 1 of hospitalization, or was variable? Explain if the samples were used from a repository bank or How patients/participants were enrolled. All patients with clinical suspicion of leptospirosis were enrolled? Are all the 112 patients with leptospirosis with kidney injury? Explain how and from where the healthy controls were selected.
- Please provide more details about the clinical characteristics of patients. For example, male/female ratio, the percentage with kidney injury, how kidney injury was determined, information about the results of diagnostic tests for leptospirosis (how many were positive for microscopy, serology and/or PCR) etc.
- Please provide information on sensitivity and specificity (95% CI) associated with the AUC for each biomarker from ROC analysis. Please determine if the difference in values of AUC for each biomarker is statistically significant.
- Discussion should be significantly improved. Authors stated several times (lines 137, 168, 179, 195) that correlation between markers “indicates involvement in kidney injury” that is speculative. Line 137 “conforming that OPN played a role in kidney disease”- this is association, not conformation. Direct experiments should be done to demonstrate direct involvement of OPN or Gal9 in development of renal injury. Discuss why OPN is increase. It is secreted by DC and macrophages and stimulates Th1 response. But normal response to Leptospira is Th2 –mediated humoral response. OPN may be a survival factor that protect from apoptosis, but major tubular injury is associated with necrosis. Explain the advantage of OPN compare to other renal markers.
Minor comments
Line 19: Abstract: Remove “t” after the word leptospirosis.
There is no method section in the abstract
Line 52: Please, “change 1 month” to “one month”.
Please keep consistency with the abbreviation for p-value in all figures. Change p<0.0001 to p<0.001. Provide description of abbreviations (*; **, ***, ****) in the figure legends. Keep this consistency through the entire document.
The information from line 39 to 43 is irrelevant for the article.
“However, we have reported three cases of leptospirosis infected in the northern part of Japan, two of which involved female patients with an average year of 80 years, indicating the wide age range of patients [3]. We have not encountered an outbreak of leptospirosis after flooding in Japan, although such outbreaks frequently occur after floods in developing countries [4].”
The expression (line 65) “reflect the disease activity well,” is very broad. Please explain and provide more detailed information.
Please, explain the expression (line 66):“opposing immunological activities”.
Line 234-236: Provide information about diagnostic tests used in the study, such as the complete name of the tests and manufacturer information.
Line 128-129: “No correlation was found with other markers (Fig.4)” Please clarify: No correlation of pFL-Gal-9 levels?
Lines 153-154, the authors said that pFL-Gal-9 could be an indicator of disease severity but was not associated with kidney injury. Can the authors explain this statement? What clinical conditions due to leptospirosis define severity of disease?
Line 162-166: It is not clear why this information is relevant for the study. Suggestion: The author can explain the association of this information with the study or remove these sentences.
Line 166-168: Can the authors explain the homeostatic roles of these molecules? It is broadly mention later in the discussion, but to improve readability, it should be explained before.
In limitations, the authors can add that they compare the results of leptospirosis patients to healthy controls. It would be interesting to compare to leptospirosis patients without kidney injury as well as to other infectious diseases or diseases with similar clinical manifestations and geographic distribution. Longitudinal studies with larger sample sizes will also be interesting to validate the results.
Explain how these two biomarkers can help in the diagnosis and disease severity in areas with similar infectious diseases since the authors indicated that these two biomarkers are also important during dengue and malaria infections.
Author Response
To the reviewer 1.
I should express my sincere appreciation for your valuable and constructive comments which helped a lot to improve the manuscripts. All the comments and replies were listed below. I hope paper is now suitable for publication to Diagnostics.
Major comments:
- The title is a little confusing since it could suggest that pFL-Gal-9 cannot be used as a biomarker of kidney injury in leptospirosis patients. The authors evaluated pFL-Gal-9 at only one-time point during infection. With the current results, the authors cannot suggest that pFL-Gal-9 is not a relevant biomarker of early kidney injury in leptospirosis, since previous studies in mice indicate that pFL-Gal-9 is elevated during acute kidney injury. Suggestion: Plasma Osteopontin Levels is associated with biochemical markers of Kidney Injury in Patients with Leptospirosis
Reply:Thank you very much for your very kind suggestions. The title was changed, accordingly, pFL-Gal-9 part in abstract was deleted.(line 33) and conclusion was changed (317-320)
- The rational to use osteoponin and galestin as biomarkers of kidney injury is not clear, especially because more common biomarkers of tubular injury (uNAG, CyC, sCr and uCLU) can well detect it. Morover correlations between uCyC/CRE, uCLU/Cr and uNAG/Cr are the highest in the st of markers (Fig.4).
Reply:Thank you for your important comments, we described this points in Introduction (lines66-60 new ref. 14-16 and discussion and discussed the possible significance of OPN in also in lepto induced CKD. (lines 259-268 new refs 28, 33, 34) and Gal-9 in glomerulonephritis (lines 302-304 Ref 15).
- Provide information about the percentage of patients with leptospirosis that develop acute tubular disease and acute kidney failure.
Reply:It was reported that 40-60 percent of patients develop AKI in newly cited Ref (6) (lines 45-46).
- Matherial and methods: Study subjects. It is not clear the type of study design that was used: Case-control study, cross-sectional study, age-matched case-control, etc. When the samples were collected: time 0, 1 of hospitalization, or was variable? Explain if the samples were used from a repository bank or How patients/participants were enrolled. All patients with clinical suspicion of leptospirosis were enrolled? Are all the 112 patients with leptospirosis with kidney injury? Explain how and from where the healthy controls were selected.
Reply: Thank you for your important questions. All the information were described in text. (lines 81-88)
- Please provide more details about the clinical characteristics of patients. For example, male/female ratio, the percentage with kidney injury, how kidney injury was determined, information about the results of diagnostic tests for leptospirosis (how many were positive for microscopy, serology and/or PCR) etc.
Reply:Male and female ration were described in Results (lines 134-135) and methods for detection of leptospira infection were described (lines 84-87). Kidney injury was determined using the increase levels of serum creanitine and 39% of the enrolled leptospirosis patients tested positive for sCr (line 220).
- Please provide information on sensitivity and specificity (95% CI) associated with the AUC for each biomarker from ROC analysis. Please determine if the difference in values of AUC for each biomarker is statistically significant.
Reply: According to your suggestions, we added table 1. (lines 149-150, 153-155) and its analytical method (line 124) Dr. Gaowa Bai who analyzed and made the table was added as co-author.(line 324)
- Discussion should be significantly improved. Authors stated several times (lines 137, 168, 179, 195) that correlation between markers “indicates involvement in kidney injury” that is speculative. Line 137 “conforming that OPN played a role in kidney disease”- this is association, not conformation. Direct experiments should be done to demonstrate direct involvement of OPN or Gal9 in development of renal injury. Discuss why OPN is increase. It is secreted by DC and macrophages and stimulates Th1 response. But normal response to Leptospira is Th2 –mediated humoral response. OPN may be a survival factor that protect from apoptosis, but major tubular injury is associated with necrosis. Explain the advantage of OPN compare to other renal markers.
Reply: Thank you very much for your very important questions on OPN and its immunobiological aspects. The description of OPN in kidney injury was softened as follows. Line 223, 247-248, 283) But line 179 in the original version mentioned only about clusterin correlation (lines 258-269). I also agree with you and believe more histo-patajologcal examination of OPN in leptospirosis would be necessary. However, it is also difficult to obtain biopsy or necropsy samples from the patients for further analysis of tubular change and OPN in our samples. We discussed on the roles of opn in lupus nephritis (Ref 33, 12)I also agree that OPN is derived from infiltrating macrophages as described in the new references ( Ref 28) and enhance Th1 mediated inflammatory response and plays a key role in apoptosis (lines 66-67, Ref 14)
We also added the important reference of Gal-9 in glomerulonephritis.
(line 67-68, ref 15) and their apoptosis inducing activity (lines 68-68,245-246,301-303 ref 16)
Minor comments
Line 19: Abstract: Remove “t” after the word leptospirosis.
Reply: “t” was removed.
There is no method section in the abstract
Reply: Method was added in the abstract (lines 21-23)
Line 52: Please, “change 1 month” to “one month”.
Reply: It was changed (line 50)
Please keep consistency with the abbreviation for p-value in all figures. Change p<0.0001 to p<0.001. Provide description of abbreviations (*; **, ***, ****) in the figure legends. Keep this consistency through the entire document.
Reply: The description was added in all figure legends. (lines 160-161,179-180, 201-203,215, 229-230)
The information from line 39 to 43 is irrelevant for the article.
Reply: It was deleted.
The expression (line 65) “reflect the disease activity well,” is very broad. Please explain and provide more detailed information.
Reply: Summary of the study were added. (lines 63-65)
Please, explain the expression (line 66):“opposing immunological activities”.
Reply; It was explained (lines 66-69)
Line 234-236: Provide information about diagnostic tests used in the study, such as the complete name of the tests and manufacturer information.
Reply: Thank you for your comments and detailed information were given.(lines 83-87)
Line 128-129: “No correlation was found with other markers (Fig.4)” Please clarify: No correlation of pFL-Gal-9 levels?
Reply: Yes. The description was corrected. (lines; 205-206)
Lines 153-154, the authors said that pFL-Gal-9 could be an indicator of disease severity but was not associated with kidney injury. Can the authors explain this statement? What clinical conditions due to leptospirosis define severity of disease?
Reply: Unfortunately, clinical data were not available in these patients, some speculations based on recent findings on Gal-9 were added.(lines 236-239, ref 24,25)
Line 162-166: It is not clear why this information is relevant for the study. Suggestion: The author can explain the association of this information with the study or remove these sentences.
Reply: To avoid the confusion, the sentences were deleted.
Line 166-168: Can the authors explain the homeostatic roles of these molecules? It is broadly mention later in the discussion, but to improve readability, it should be explained before.
Reply: To avoid the confusion, the sentences were deleted.
- In limitations, the authors can add that they compare the results of leptospirosis patients to healthy controls. It would be interesting to compare to leptospirosis patients without kidney injury as well as to other infectious diseases or diseases with similar clinical manifestations and geographic distribution. Longitudinal studies with larger sample sizes will also be interesting to validate the results.
Reply: Thank you for your encouragements, we agree with your criticisms and it was described. (lines 315)
Explain how these two biomarkers can help in the diagnosis and disease severity in areas with similar infectious diseases since the authors indicated that these two biomarkers are also important during dengue and malaria infections.
Reply: It is evident that these biomarkers do not reflect pathogens significances of OPN in CKD development were described (267-2680, ref 34). We also added possible beneficial effect of Gal-9 (lines 245-246, 301-303 ref 15)
Reviewer 2 Report
-How were patients with Leptospirosis chosen? How leptospirosis was diagnosed? How long did these patient have leptospirosis? Acute phrase, subacute, or resolution already, the findings are surely different among different phrase. Need to be more clear, all patients during what period of time? Currently, the described selection process is unclear and can lead to selection bias.
-The number of Institutional Review Board (IRB) approval for this study should be provided with IRB approval statement.
-The authors should discuss what the clinical utility of these findings could be - if any.
-There should be more discussion of a possible mechanism
-Explain what sort of future study might take us closer to a clinical utility
-Please reduce self-citations. The investigators cited their works at least 11 citations, and some of them are not necessary. One of the evaluation scores is making sure there is no self-citation.
-Figure 1, Figure 3 and Figure 5 need to increase in size. They are very difficult to assess.
Author Response
To the reviewer 2
I should express my sincere appreciation for your valuable and constructive comments which helped a lot to improve the manuscripts. All the comments and replies were listed below. I hope paper is now suitable for publication to Diagnostics.
Toshio
- Comments and Suggestions for Authors
- -How were patients with Leptospirosis chosen? How leptospirosis was diagnosed? How long did these patient have leptospirosis? Acute phrase, subacute, or resolution already, the findings are surely different among different phrase. Need to be more clear, all patients during what period of time? Currently, the described selection process is unclear and can lead to selection bias.
Reply. Thank you for the very important questions. We described as much as we could. These patients came to the hospitals after flood assuming that they are acute patients, unfortunately follow up clinical records were not available. It was described (lines 81-87, 309-315)
- -The number of Institutional Review Board (IRB) approval for this study should be provided with IRB approval statement.
Reply: It is stated in lines (lines 128-129).
- -The authors should discuss what the clinical utility of these findings could be - if any.
Reply: possible application of FL-OPN to monitor if they could be a marker for CKD were described (lines 264-268 refs 33,34). Also possible beneficial effect of Gal-9 were described (lines 245-246, 301-303, ref 15)
- -There should be more discussion of a possible mechanism.
Reply: In introduction, Th-1 inducing and apoptosis modulating function of OPN and Th1 suppressing and apoptosis inducing function of Gal-9 were introduced and clarified that OPN and Gal-9 have opposing biological function (lines 66-69) (refs 14-16) and clarified the inverse correlations of OPN and Gal-9 are important (line 292) We discussed infiltration of macrophages in leptospirosis and proposed that the macrophages may be the source of OPN (lines 264-268, ref 28). Degradation of OPN and different ELISA system which detects both FL-OPN and degraded OPN in lupus nephritis was introduced (lines 259-264, ref 12,33). Finally possible application of OPN to monitor CKD development were described (lines 264-268 refs 33,34). The newly proposed function as immune check point inhibitors of Gal-9 was introduced and implied that their accelerated degradation may cause cytokine storm (lines 236-239). Possible beneficial effect of Gal-9 were also described and proposed the effect could explain the lack of correlation of Gal-9 levels with kidney toxin marker (lines 245-246, 301-303, ref 15).
- -Explain what sort of future study might take us closer to a clinical utility.
Reply: Longitudinal studies with larger sample sizes with other clinical parameters , should be conducted to validate the results. These experiments would hep to understand if OPN is useful as a monitoring development of CKD. It is possible to identify Gal-9 levels indicate the risk of cytokine storm or could implicate theor tole in amelioration,
- -Please reduce self-citations. The investigators cited their works at least 11 citations, and some of them are not necessary. One of the evaluation scores is making sure there is no self-citation.
Reply: Thank you for your suggestions we deleted two references.
- -Figure 1, Figure 3 and Figure 5 need to increase in size. They are very difficult to assess.
Reply: The size of figures were increased.
Round 2
Reviewer 1 Report
The authors have significantly improved the manuscript based on the comments. The manuscript can be accepted for publication.Reviewer 2 Report
The investigators added new author, please help make sure they include the authorship change form and declare the contributions appropriately.